# Swine Colibacillosis: Global Epidemiologic and Antimicrobial Scenario

**DOI:** 10.3390/antibiotics12040682

**Published:** 2023-03-30

**Authors:** Maria Margarida Barros, Joana Castro, Daniela Araújo, Ana Maria Campos, Ricardo Oliveira, Sónia Silva, Divanildo Outor-Monteiro, Carina Almeida

**Affiliations:** 1I.P—National Institute for Agrarian and Veterinariay Research (INIAV), Rua dos Lagidos, 4485-655 Vila do Conde, Portugal; margarida.barros@iniav.pt (M.M.B.); joana.castro@iniav.pt (J.C.); daniela.araujo@iniav.pt (D.A.); anamaria.campos@iniav.pt (A.M.C.); ricardo.oliveira@iniav.pt (R.O.); sonia.silva@iniav.pt (S.S.); 2Veterinary and Animal Research Centre (CECAV), University of Trás-os-Montes and Alto Douro, 5000-801 Vila Real, Portugal; divanildo@utad.pt; 3LEPABE—Laboratory for Process Engineering, Environment, Biotechnology and Energy, Faculty of Engineering, University of Porto, Rua Dr. Roberto Frias, 4200-465 Porto, Portugal; 4ALiCE—Associate Laboratory in Chemical Engineering, Faculty of Engineering, University of Porto, Rua Dr. Roberto Frias, 4200-465 Porto, Portugal; 5Centre of Biological Engineering, University of Minho, 4710-057 Braga, Portugal

**Keywords:** swine colibacillosis, AMR bacteria, *E. coli* pathotypes, prevalence, epidemiology

## Abstract

Swine pathogenic infection caused by *Escherichia coli*, known as swine colibacillosis, represents an epidemiological challenge not only for animal husbandry but also for health authorities. To note, virulent *E. coli* strains might be transmitted, and also cause disease, in humans. In the last decades, diverse successful multidrug-resistant strains have been detected, mainly due to the growing selective pressure of antibiotic use, in which animal practices have played a relevant role. In fact, according to the different features and particular virulence factor combination, there are four different pathotypes of *E. coli* that can cause illness in swine: enterotoxigenic *E. coli* (ETEC), Shiga toxin-producing *E. coli* (STEC) that comprises edema disease *E. coli* (EDEC) and enterohemorrhagic *E. coli* (EHEC), enteropathogenic *E. coli* (EPEC), and extraintestinal pathogenic *E. coli* (ExPEC). Nevertheless, the most relevant pathotype in a colibacillosis scenario is ETEC, responsible for neonatal and postweaning diarrhea (PWD), in which some ETEC strains present enhanced fitness and pathogenicity. To explore the distribution of pathogenic ETEC in swine farms and their diversity, resistance, and virulence profiles, this review summarizes the most relevant works on these subjects over the past 10 years and discusses the importance of these bacteria as zoonotic agents.

## 1. Introduction

Porcine infection caused by *Escherichia coli* (*E. coli*), so-called swine colibacillosis, is responsible for a wide range of problems, such as neonatal diarrhea, post-weaning diarrhea (PWD), edema disease (ED), septicemia, polyserositis, coliform mastitis, and urinary tract infection [1]. Among the huge diversity, certain strains of *E. coli*, named enterotoxigenic *E. coli* (ETEC), are able to cause intestinal disease, which results in neonatal diarrhea, PWD, and ED. These porcine infections are the most threatening for the swine industry worldwide due to significant economic losses associated with morbidity, mortality, decreased weight gain, the rising cost of treatments, vaccinations, and feed supplements [1,2,3].

PWD and ED may occur separately or together either in an individual outbreak or in the same pig [1]. Within 2–3 weeks after weaning, the piglets are more susceptible to microbial infections, owing to the existence of an immature immune system associated with sow milk removal and resulting from the interruption of the nutritive intake of immunoglobulin present in the milk [2,4]. Therefore, this period is crucial and usually associated with the most severe form of enteric *E. coli* infection, manifested by sudden death or severe diarrhea [1].

As an effort to promote health and growth performance, diverse approaches have been used to prevent and treat swine colibacillosis, with antibiotics being the most commonly used strategy [2,4]. Consequently, due to the growing selective pressure of antibiotic use to treat these *E. coli* infections, the emergence of the antimicrobial resistance (AMR) phenomenon has limited treatment options for pig producers and an increased public health concern because of the potential transfer of AMR genetic determinants directly by contact and indirectly into the food chain, water, and manure, among others [1,5]. It is important to note that *E. coli* has a great capacity to acquire resistance genes, mainly through horizontal gene transfer [6], in which the mobile genetic elements, such as plasmids, transposons, and gene cassettes in class 1 and class 2 integrons, seem to play a main role in the dissemination [5]. Furthermore, *E. coli* behaves as a donor and as a recipient of resistance genes and thus can exchange those genes with other bacteria and act as a reservoir of AMR genes [5]. Accordingly, extended-spectrum β-lactamases, carbapenemases, 16S rRNA methylases, plasmid-mediated quinolone resistance (PMQR) genes, and *mcr* genes constitute the most problematic genetic determinant classes of AMR in *E. coli* [5].

Hence, this review assembled diverse studies of importance regarding infections caused by ETEC and their epidemiologic and antimicrobial consequences at the global level during the past 10 years to bring useful and organized information about the distribution, diversity, resistance, and virulence profiles according to the pathogenic serotypes expressed by ETEC in swine farms.

## 2. Etiology

According to the taxonomy, the German pediatrician Theodor Escherich (1857–1911) gave the origin to the name of the genus *Escherichia*. This genus belongs to the family of *Enterobacteriaceae*, which contemplates the Gram-negative facultatively anaerobic rods, where the species *Escherichia coli* fits since they are Gram-negative, peritrichously flagellated rods of variable length and a diameter of about 1 μm [1].

Over the decades, the subdivision of species into types has been carried out by the development of several classification systems. Among these, serotyping (described in Table 1) is a recognized typing system to classify *E. coli* strains [1]. Nevertheless, since certain porcine pathogenic *E. coli* belong to a limited number of serotypes, this method is less used today for diagnostic purposes [1]. Thus, the serotyping technique has been substituted by the direct detection of genes coding for bacterial determinants involved in their pathogenesis, called virulence factors. Therefore, the term pathotype is applied to the classification of *E. coli* typologies according to the combinations of virulence factors [3].

Common *E. coli* pathotypes include Shiga toxin-producing *E. coli* (STEC) that contains two groups, edema disease *E. coli* (EDEC) and enterohemorrhagic *E. coli* (EHEC), enteropathogenic *E. coli* (EPEC), and extraintestinal pathogenic *E. coli* (ExPEC), with the most relevant *E. coli* pathotype in porcine, the enterotoxigenic *E. coli* (ETEC) [1]. It is important to highlight that EHEC and EPEC pathotypes are associated with the “attaching and effacing” (A/E) lesion development [1]. Nevertheless, EPEC is found in pigs with PWD, whereas EHEC is highly pathogenic in humans, and some zoonoses of this pathotype are sporadically recovered [1]. Therefore, this review is focused on ETEC due to its high pathogenicity in porcine, being the one responsible for the most cases, in number and severity, of swine enteric colibacillosis. Besides that, ETEC was classified as the most important multi-drug resistant (MDR) bacteria in pig production by the European Food Safety Authority (EFSA) [7]. Hence, in the following subchapter, the mechanisms of virulence and the pathogenesis for ETEC are detailed.

### 2.1. ETEC Virulence Factors and Their Impact on Trigger Colibacillosis Infection

*E. coli* is both a harmless commensal bacterium in the intestines of several mammals, as well as a dangerous pathogen [5]. Still, a small proportion of strains are pathogenic and can cause severe to life-threatening intestinal and extra-intestinal infections in humans and animals [5,7,8,9,10]. The pathogenicity of the strains is characterized by the presence of certain virulence factor combinations in particular adhesins and toxin secretions. As it has been described, the role of adhesins and surface proteins called fimbriae is to enable the adherence of ETEC to specific receptors on the brush borders of the small intestine’s enterocytes [8]. Regarding fimbriae, there are five common antigenically different types found in pigs: F4 (K88), F5 (K99), F41, F6 (987P), and F18 [11]. The first four fimbria types are responsible for mediating adhesion in neonates, while F18 is not associated with neonatal colibacillosis; however, it is common in postweaning colibacillosis as is F4. It is also important to highlight that hemolysis is a common trait for pathogenic F4 and F18 isolates [4]. The adhesion of ETEC through fimbriae conducts the release of toxins inside the epithelial cell that promote the secretion of water and electrolytes into the intestinal lumen [1,2,8]. ETEC constitutes the most relevant and pathogenic strains in porcine, where the following groups of toxins are produced, namely the two major classes of enterotoxins, heat-stable toxin (ST) and heat-labile toxin (LT), as well as the enteroaggregative heat-stable toxin 1 (EAST1), as described in Table 1. STs are divided into Sta (also nominated STI, ST1, or StaP) and STb (also nominated STII or ST2) [1,2]. Similarly, LT toxins are divided into two groups, LTI and LTII. On the other hand, EAST1 is widespread among porcine ETEC, but its role in this illness remains controversial [1]. It is important to note that EAST1 alone does not seem capable of developing the disease; however, together with LT, it takes effect [1,2]. Additionally, it is important to note that Shiga toxin (Stx or VT), namely the Shiga toxin type 2e (Stx2e), *E. coli* is also found in some ETEC strains, being commonly found in ETEC strains expressing LT and/or ST toxins [11]. This toxin is a causative factor of edema in swine, and it has been also associated with diarrhea commonly found in colibacillosis [1,11].

Regarding the route of contamination, ETEC is firstly ingested through the oral route and then passes through the stomach. When it reaches the intestine, in the presence of suitable environmental conditions, ETEC proliferates and causes disease [1,8]. At this point, they colonize the small intestine following the attachment of fimbria adhesins to specific receptors present on the epithelium as well as in the mucus coating the epithelium of the small intestine (see Figure 1). This promotes the production and release of enterotoxins, as mentioned above, inside the epithelial cell that stimulates the secretion of water and electrolytes into the intestinal lumen, which leads to diarrhea, weight loss, and possibly death [1,3,8]. To note, the combinations of different ETEC virulence factors (adhesins and toxins) are associated with the development of different diseases, namely neonatal diarrhea and PWD, as shown in Table 1.

**Table 1 antibiotics-12-00682-t001:** Virulence factors of enterotoxigenic *Escherichia coli* (ETEC) associated with swine enteric colibacillosis.

Adhesins	Toxins	Serotypes	Disease
F5, F6, F41	STa	O8, O9, O20, O64, O101	Neonatal diarrhea
F4	STa, STb, LT, EAST1, α-hemolysin ^b^	O8, O138, O141, O145, O147, O149, O157	Neonatal diarrhea Diarrhea in young pigs preweaning
F4, AIDA ^a^, unknown	STa, STb, LT, EAST1, α-hemolysin ^b^	O8, O138, O139, O141, O147, O149, O157	PWD
F18, AIDA ^a^	STa, STb, LT, Stx (or VT) ^c^, EAST1, α-hemolysin ^b^	O8, O138, O139, O141, O147, O149, O157	PWD

^a^ AIDA is a non-fimbrial adhesin involved in diffuse adherence; nevertheless, the mechanism of AIDA in swine colibacillosis is not yet elucidated. However, this non-fimbrial adhesin has been associated with ETEC strains from weaned pigs with PWD [2]. ^b^ α-hemolysin is a pore-forming cytolysin associated with ETEC strains that cause diarrhea in animals [12]. ^c^ Shiga toxins (Stx or VT) are cytotoxins produced by Shiga-toxin-producing *E. coli* (STEC), and in swine, the most important STECs are those that cause edema disease (ED) [1].

#### 2.1.1. Neonatal ETEC

Neonatal diarrhea caused by ETEC is only associated with STa and might have one or more of the fimbriae associated, including F4 (K88), F5 (K99), F6 (987P), and F41 [1]. Of note, F4 is usually related to the colonization of the length of the jejunum and ileum, whereas F5, F6, and F41 mostly colonize the posterior jejunum and ileum [2]. Recently, Dubreuil and colleagues reviewed the prevalence of F4-, F5-, and F6-fimbriated ETEC from neonatal diarrhea and concluded that they exhibited both temporal and geographic variations [13]. This illness usually occurs in the first four days after the piglet is born and is characterized by whitish-yellow diarrhea, with a watery or creamy consistency, in large quantities [2]. Once ETEC has adhered to the epithelium, it binds to specific receptors in the apical region of the epithelial cells in the jejunal region and begins to produce the enterotoxin Sta, culminating in the hypersecretion of electrolytes and fluids in the small intestine. Eventually, if the large intestine is unable to reabsorb these excess fluids, the piglet enters a state of dehydration (and in more severe cases, metabolic acidosis) and eventually dies [1].

#### 2.1.2. Postweaning ETEC

The enterotoxins STa, STb, LT, and EAST-1 are typically produced individually or together in the ETECs that cause diarrhea in postweaning or older suckling pigs [1]. The fimbriae involved in this disease are mainly F4 and F18, and both possess several variant subtypes based on antigenic differences. In this respect, F4 and F18 evidenced the following subtypes, ab, ac, and ad and ab and ac, respectively [2]. Post-weaning diarrhea usually appears 2 to 3 weeks after weaning and is characterized by the appearance of watery diarrhea that can vary from yellowish-grey or a slightly pinkish color, which lasts for up to a week [1]. The ETEC responsible for this condition generally colonizes the duodenal and jejunal portions of the small intestine, thus inducing the hypersecretion of fluids, very similar to that described in neonatal diarrhea [1].

## 3. Global Epidemiology of Swine Enteric Colibacillosis: Prevalence, Diversity, and Outbreaks

Enteric colibacillosis in swine is related to high morbidity and mortality [1,14]. It has been reported that mortality can reach up to 70% in neonatal piglets with severe watery diarrhea, 1.5–2% in post-weaned and/or grow-finish pigs with moderate diarrhea, and up to 25% in untreated pigs with severe to moderate diarrhea [1,7]. In fact, this remarkable swine infection is widespread, taking place in both industrialized and developing countries and in temperate, subtropical, and tropical climates [1,14].

It is important to note that, as mentioned briefly above, this infection requires the presence, by ingestion, of ETEC and specific predisposing environmental conditions and host factors. Thus, these strains proliferate in the intestine and cause illness due to specific virulence factors, as reported in Section 2.1. The degree of ETEC colonization determines the occurrence of the disease [2]. Interestingly, it was already shown that ETEC strains were present in 16.6% of non-diarrhoeic pigs during the piglets’ suckling period, 66% in the nursery phase, and 17.3% in the finisher population. Furthermore, ETEC strains can be shed in the feces of healthy pigs [2,7].

ETECs can be found in fecal-contaminated feed, water, and soil and the environment of the pig barn. Long survival times in the environment are achieved by low temperatures and enough moisture, among other factors [1]. In slurry samples, a porcine ETEC O139:K82 strain remained viable for more than 11 weeks [1]. The spread of pathogenic *E. coli* is supposed to mainly occur via other pigs and contaminated barn environments. In addition, other transmission modes, namely via aerosols, have been reported [1]. Importantly, it was shown that airborne transmission between pigs in wire cages 1.5 m apart was repeatedly observed in transmission experiments with an F4-ETEC strain [1]. In addition, other possible modes of transmission are contaminated feed and water, contaminated trucks that transport pigs, and possibly other animal species. As a result of this transmission cycle, the same strain is usually found in many sick pigs and often in consecutive batches of pigs. To control the transmission of this infection, it is necessary to use strict hygienic measures [1] since routine cleaning and disinfection are usually insufficient to break the cycle of infection by ETEC [15].

Complete information on the prevalence is scarce, which makes the comparison between countries difficult. However, previous analysis already suggested some differences, as shown in Table 2.

Regarding the diversity and heterogeneity of ETEC, it is important to highlight that ETEC populations in pig fecal microbiota and in the farm environment are very dynamic and show high levels of diversity [2,25]. Numbers in the large intestine average around 10^7^ colony-forming units (CFU)/g of contents; nevertheless, *E. coli* contributes less than 1% to the total bacterial count [1].

A recent study across Europe concerning ETEC pathotypes demonstrated a higher prevalence of F4 compared to F18 isolates in Belgium and the Netherlands, France, and Italy, as can be seen in Table 2 [18]. In contrast, in Spain, a different tendency was observed, with F18 being the prevalent adhesin [23]. In fact, the reported association of F18 isolates with PWD in other countries varied widely, as can be seen in Table 3, from 15.4% in Australia [26], 35% in Slovakia [27], 39.3% in Denmark [28], and 53% in the United States [29] to 61.9% in Poland [30]. Furthermore, a high prevalence of F18 was reported in Japan (62.9%), where, in addition, most isolates carried the *stx2e* gene (60.1%), which describes a very different pathogenic profile concerning other geographic areas and represents a high risk to swine production [31]. It is important to highlight that, in Spain, around 10% of Stx2e-positive isolates were found [23] similar to what was reported in other European countries [18].

It is also remarkable that, in Spain, according to García-Meniño and colleagues, the most common virulence profiles within each pathotype were LT, STb, and F4 and LT, STa, STb, and F18 (37.3% and 18.6% of the 161 ETEC isolates, respectively) [22]. According to the area of study, there is a divergence in the combination of virulence factors harboring the ETEC; in 2010, LT, STb, and F18 were the most predominant genotypes in the United States, for instance [29]. As expected, it was also previously described that the distribution of enterotoxins/fimbriae can also vary over time in a region [2,21]. An example is the United States, where, in 2001–2002, LT, STb, and F4 were the most prevalent genotype [32], which differ from the previous study [29].

It is also important to mention that LT and ST are well-known enterotoxins responsible for the diarrhea symptom, while it has been proposed that Stx2e is responsible for the severe neurological damage observed in swine edema disease [31]. Although enough data do not exist to assess the differences in disease severity between Stx-producing ETEC and other virulence factor-containing isolates, it has been reported that swine infected with *E. coli*-producing enterotoxins and Stx2e commonly exhibit diarrhea as an initial symptom, which is followed by lethal neurological symptoms [33].

**Table 3 antibiotics-12-00682-t003:** The prevalence of the most known fimbriae and toxins in ETEC strain isolates from swine.

Country (n = Number of Isolates)	Percentage (Number) of Positive Isolates (%)									Reference
	Fimbriae					Toxins				
	F4	F5	F6	F18	F41	LT	STa	STb	Stx2e	
Australia (n = 104)	38.5–96.3	-	-	0–15.4	-	62.1–92.3	64.8–92.3	83.7–100	-	[26]
Belgium and The Netherlands (n = 100)	51.0	1.0	1.0	42.0	-	14.0	22.0	30.0	5.0	[18]
Denmark (n = 219)	44.7	-	0.9	39.3	-	61.6	26.5	77.6	-	[28]
France (n = 91)	47.3	-	-	35.2	-	45.1	40.7	76.9	19.8	[18]
Germany (n = 64)	14.1	-	-	14.1	-	9.4	26.6	57.8	3.1	[18]
Italy (n = 84)	59.3	1.2	1.2	38.1	1.2	56.0	63.1	71.4	9.5	[18]
Poland (n = 40)	22.5	-	-	61.9	-	22.5	72.5	77.5	17.5	[30]
Spain (n = 181)	38.2	4.8	1.1	43.5	2.7	66.1	50.5	74.7	13.5	[22]
Spain ^a^ (n = 277)	27.7–40.5	16.7	11.9	51.5	16.7	-	-	-	10	[25]
Slovakia (n = 101)	19	0.9	5	35	0.9	20	26	46	5	[27]
Uganda ^b^ (n = 83)	8.4	-	-	-	-	-	1.2	26.5	2.4	[34]
United States (n = 175)	41.7	-	-	53.1	-	52.6	38.2	96	-	[29]
Zimbabwe (n = 1984)	28.4	22.3	1.5	25.4	22.3	50	73	16	27	[35]

^a^ All isolates were positive for genes encoding enterotoxins (STa and/or STa and/or STb). ^b^ A prevalence of 16% of AIDA was detected. Note: “-”, means not found.

## 4. Antimicrobial Prevalence in Enteric Colibacillosis Treatment

The high morbidity and mortality rates are not the only problems associated with enteric colibacillosis in swine production but also the cost associated with its treatment with antibiotics [2]. Moreover, the spread of AMR determinants combined with the decrease in the available antimicrobial treatments is currently a global problem. Normally, antibiotics should be administered only to pigs that show clinical signs of colibacillosis; however, when the mortality increases in the farm production, the prophylactic treatment is applied to all animals [22]. However, the use of antimicrobials for the growth promotion of food animals has been banned in several countries [36]. It is noteworthy that, in Europe, according to Regulation (EU) 2019/61 on Veterinary Medicines and Regulation (EU) 2019/4 on Medicated Feed, antibiotics shall not be applied routinely, nor used for prophylaxis, unless in exceptional cases. It should only be applied for metaphylaxis when the risk of spreading infection is very high and there are no other options. Similarly, in the USA, since 2017, growth promotion uses of medically important (to human health) antibiotics are not allowed. Only therapeutic use (treatment, control, prevention) for a specific animal health condition is allowed under the direction of a veterinarian [37,38]. Additionally, in Brazil, since 1998, the use of several antimicrobial classes as growth promoters is prohibited, and recently, in 2016, colistin, a last-resort treatment for multidrug-resistant Gram-negative infections, was also banned [39].

In fact, it has been reported that the level of AMR in the gut microbiota increases with the number of antimicrobials used [40]. Due to the emergence of antimicrobial-resistant bacteria and the spread of AMR genes, antimicrobial resistance has become a global problem in the swine industry [5]. To overcome this problem, several countries have been monitoring AMR [5].

One factor that contributes to promoting the AMR phenomenon is the fact that *E. coli* presents a high capacity to acquire and pass antimicrobial resistance genes via horizontal gene transfer [5]. In fact, *E. coli* isolates revealed an alarming scenario with high resistance to a different antimicrobial class, as is the case of penicillins, aminoglycosides, tetracyclines, sulphonamides, fluoroquinolones, and phenicols [2]. In 2019, *E. coli* was considered one of the major pathogens responsible for the deaths associated with AMR [41]. Therefore, there is an increasing trend in the detection of AMR among ETEC from pigs with enteric colibacillosis [2]. Table 4 presents several studies from different countries on the resistance profile to different antimicrobial classes of *E. coli* isolated from swine with the disease. In Spain, ETEC isolates from a swine farm presented high levels of resistance to antibiotics commonly used for the treatment, such as in the case of ampicillin (Table 4) [22,23]. Similar values of resistance to ampicillin were observed in several regions in China [42,43,44,45]. Tetracycline is another antibiotic widely used in veterinary medicine that has been reported in several studies with a huge AMR in different parts of the world. A recent study developed in Denmark revealed a percentage of around 57% resistance to tetracycline of 90 ETEC isolates from pigs [46]. Furthermore, in the same study, antibiotics of other classes presented a resistance higher than 50%, namely spectinomycin, streptomycin, sulfamethoxazole, and trimethoprim (Table 4).

The variation of resistance in ETEC isolates in different countries emphasizes the importance of performing antimicrobial susceptibility testing in farm productions to select the correct antimicrobial agent for the treatment of ETEC. Moreover, it is not allowed in several countries to use antimicrobials for growth production since healthy pigs can serve as reservoirs for resistant *E. coli* and resistant bacteria can be transferred from the animals to humans by direct contact or by the food chain or indirectly through the environment [5,47].

**Table 4 antibiotics-12-00682-t004:** The prevalence of antimicrobial resistance among *E. coli* isolates from swine over the world.

Antimicrobial Class/Other Designations	Antimicrobial Agents	% Resistant Rates (n = Swine Isolates)	Country/City	Year/Time Range of the Study	Reference
Penicillins	Ampicillin	85.9 (n = 608)	China/Shanghai	2009–2021	[48]
		75.4 (n = 481)	Spain/Lugo	2006–2016	[23]
		71.9 (n = 694)	Austria/Vienna	2016–2018	[49]
		84.8 (n = 455)	China/Beijing	2014–2016	[42]
		27.9 (n = 129)	China/Tibet	2012	[43]
		60.7 (n = 89)	Denmark/Frederiksberg C	2014	[46]
		81.4 (n = 161)	Spain/Lugo	2005–2017	[22]
		60.86 (n = 23)	Bangladesh/Tangail	2018	[50]
		86.4 (n = 118)	Korea	2016–2017	[40]
		48.3 (n = 90)	Denmark	2018–2019	[51]
		89.1 (n = 55)	United States	2013–2014	[44]
		34.5 (n = 168)	China/Shenzhen	2009–2014	[45]
	Ampicillin-sulbactam	64.6 (n = 481)	Spain/Lugo	2006–2016	[23]
	Ticarcillin	73.8 (n = 481)	Spain/Lugo	2006–2016	[23]
		81.4 (n = 161)	Spain/Lugo	2005–2017	[22]
β-lactam combination agents	Amoxicillin/clavulanic acid	42.3 (n = 608)	China/Shanghai	2009–2021	[48]
		84.63 (n = 455)	China/Beijing	2014–2016	[42]
		11.76 (n = 135)	Santa Catarina/ Brazil	2016–2017	[52]
		33.5 (n = 161)	Spain/Lugo	2005–2017	[22]
		82.6 (n = 23)	Bangladesh/Tangail	2018	[50]
		5.1 (n = 118)	Korea	2016–2017	[40]
		1.1 (n = 90)	Denmark	2018–2019	[51]
		9.5 (n = 168)	China/Shenzhen	2009–2014	[45]
	Ampicillin/ sulbactam	70.8 (n = 161)	Spain/Lugo	2005–2017	[22]
		5 (n = 168)	China/Shenzhen	2009–2014	[45]
Penicillins + *β*-lactamase inhibitors	Piperacillin/ tazobactam	0.6 (n = 161)	Spain/Lugo	2005–2017	[22]
Cephalosporins	Ceftiofur	22.5 (n = 608)	China/Shanghai	2009–2021	[48]
		52.63 (n = 455)	China/Beijing	2014–2016	[42]
		25 (n = 135)	Santa Catarina/ Brazil	2016–2017	[52]
		10.9 (n = 129)	China/Tibet	2012	[43]
		25.5 (n = 55)	United States	2013–2014	[44]
	Cefepime	9.2 (n = 481)	Spain/Lugo	2006–2016	[23]
		7.5 (n = 161)	Spain/Lugo	2005–2017	[22]
		2.5 (n = 118)	Korea	2016–2017	[40]
		4.2 (n = 168)	China/Shenzhen	2009–2014	[45]
	Cefazolin	60.82 (n = 455)	China/Beijing	2014–2016	[42]
		10.6 (n = 161)	Spain/Lugo	2005–2017	[22]
		10.2 (n = 118)	Korea	2016–2017	[40]
	Cefuroxime	8.7 (n = 161)	Spain/Lugo	2005–2017	[22]
	Cefotaxime	10.6 (n = 161)	Spain/Lugo	2005–2017	[22]
		9.1 (n = 168)	China/Shenzhen	2009–2014	[45]
	Ceftazidime	5 (n = 161)	Spain/Lugo	2005–2017	[22]
		3 (n = 168)	China/Shenzhen	2009–2014	[45]
	Cephalothin	64.4 (n = 118)	Korea	2016–2017	[40]
		41.7 (n = 168)	China/Shenzhen	2009–2014	[45]
	Cefoxitin	3.4 (n = 118)	Korea	2016–2017	[40]
		1.8 (n = 168)	China/Shenzhen	2009–2014	[45]
	Ceftriaxone	6 (n = 168)	China/Shenzhen	2009–2014	[45]
Carbapenems	Ceftazidime	1.9 (n = 608)	China/Shanghai	2009–2021	[48]
		1.5 (n = 481)	Spain/Lugo	2006–2016	[23]
		5.9 (n = 694)	Austria/Vienna	2016–2018	[49]
	Meropenem	0.3 (n = 608)	China/Shanghai	2009–2021	[48]
Aminoglycosides	Kanamycin	63.74 (n = 455)	China/Beijing	2014–2016	[42]
		3.6 (n = 168)	China/Shenzhen	2009–2014	[45]
	Spectinomycin	65.7 (n = 608)	China/Shanghai	2009–2021	[48]
		2.3 (n = 129)	China/Tibet	2012	[43]
		18 (n = 89)	Denmark/Frederiksberg C	2014	[46]
		43.6 (n = 55)	United States	2013–2014	[44]
		55.6 (n = 90)	Denmark	2018–2019	[51]
	Gentamicin	37.2 (n = 608)	China/Shanghai	2009–2021	[48]
		47.7 (n = 481)	Spain/Lugo	2006–2016	[23]
		7.7 (n = 694)	Austria/Vienna	2016–2018	[49]
		57.31 (n = 455)	China/Beijing	2014–2016	[42]
		32.35 (n = 135)	Santa Catarina/ Brazil	2016–2017	[52]
		6.9 (n = 129)	China/Tibet	2012	[43]
		14.6 (n = 89)	Denmark/Frederiksberg C	2014	[46]
		58.4 (n = 161)	Spain/Lugo	2005–2017	[22]
		36.4 (n = 118)	Korea	2016–2017	[40]
		32.7 (n = 55)	United States	2013–2014	[44]
		6.7 (n = 90)	Denmark	2018–2019	[51]
		5.4 (n = 168)	China/Shenzhen	2009–2014	[45]
	Tobramycin	47.7 (n = 481)	Spain/Lugo	2006–2016	[23]
		6.2 (n = 694)	Austria/Vienna	2016–2018	[49]
		54.7 (n = 161)	Spain/Lugo	2005–2017	[22]
	Streptomycin	40.35 (n = 455)	China/Beijing	2014–2016	[42]
		16.2 (n = 129)	China/Tibet	2012	[43]
		29.2 (n = 89)	Denmark/Frederiksberg C	2014	[46]
		86.4 (n = 118)	Korea	2016–2017	[40]
		68.9 (n = 90)	Denmark	2018–2019	[51]
		18.5 (n = 168)	China/Shenzhen	2009–2014	[45]
	Amikacin	15.2 (n = 455)	China/Beijing	2014–2016	[42]
		1.2 (n = 168)	China/Shenzhen	2009–2014	[45]
	Apramycin	14.6 (n = 89)	Denmark/Frederiksberg C	2014	[46]
		8.9 (n = 90)	Denmark	2018–2019	[51]
	Neomycin	50 (n = 118)	Korea	2016–2017	[40]
		49.1 (n = 55)	United States	2013–2014	[44]
		25.6 (n = 90)	Denmark	2018–2019	[51]
Tetracyclines	Doxycycline	85.9 (n = 608)	China/Shanghai	2009–2021	[48]
		62.7 (n = 161)	Spain/Lugo	2005–2017	[22]
	Tetracycline	91.6 (n = 608)	China/Shanghai	2009–2021	[48]
		67.7 (n = 694)	Austria/Vienna	2016–2018	[49]
		83.63 (n = 455)	China/Beijing	2014–2016	[42]
		40.4 (n = 129)	China/Tibet	2012	[43]
		47.2 (n = 89)	Denmark/Frederiksberg C	2014	[46]
		65.21 (n = 23)	Bangladesh/Tangail	2018	[50]
		86.4 (n = 118)	Korea	2016–2017	[40]
		56.7 (n = 90)	Denmark	2018–2019	[51]
		21.4 (n = 168)	China/Shenzhen	2009–2014	[45]
	Minocycline	41.5 (n = 481)	Spain/Lugo	2006–2016	[23]
		52.2 (n = 161)	Spain/Lugo	2005–2017	[22]
	Chlortetracycline	80 (n = 55)	United States	2013–2014	[44]
	Oxytetracycline	94.5 (n = 55)	United States	2013–2014	[44]
Sulfonamides	Sulfisoxazole	85.4 (n = 608)	China/Shanghai	2009–2021	[48]
	Sulphaamethoxazole	75.2 (n = 608)	China/Shanghai	2009–2021	[48]
		69.7 (n = 89)	Denmark/Frederiksberg C	2014	[46]
		67.8 (n = 90)	Denmark	2018–2019	[51]
	Sulfadimethoxine	61.8 (n = 55)	United States	2013–2014	[44]
Fluoroquinolones	Enrofloxacin	41.3 (n = 608)	China/Shanghai	2009–2021	[48]
		72.51135(n = 455)	China/Beijing	2014–2016	[42]
		54.41 (n = 135)	Santa Catarina/Brazil	2016–2017	[52]
		58.2 (n = 55)	United States	2013–2014	[44]
	Ofloxacin	39 (n = 608)	China/Shanghai	2009–2021	[48]
	Ciprofloxacin	61.5 (n = 161)	Spain/Lugo	2005–2017	[22]
		26.3 (n = 118)	Korea	2016–2017	[40]
		9.8 (n = 41, farm 1); 8.8% (n = 34, farm 2); 21.7% (n = 23, farm 3); 39.6% (n = 48, farm 4); 3.4% (n = 58, farm 5); 50% (n = 24, farm 6); 70% (n = 10, farm 7)	Germany Mecklenburg–Western Pomerania	2018	[53]
		12.3 (n = 481)	Spain/Lugo	2006–2016	[23]
		16.4 (n = 694)	Austria/Vienna	2016–2018	[49]
		60.82 (n = 455)	China/Beijing	2014–2016	[42]
		7.8 (n = 129)	China/Tibet	2012	[43]
		47.82 (n = 23)	Bangladesh/Tangail	2018	[50]
		3.6 (n = 168)	China/Shenzhen	2009–2014	[45]
	Levofloxacin	55.3 (n = 161)	Spain/Lugo	2005–2017	[22]
		3.6 (n = 168)	China/Shenzhen	2009–2014	[45]
Polymyxins	Colistin	21.9 (n = 608)	China/Shanghai	2009–2021	[48]
		76.4 (n = 161)	Spain/Lugo	2005–2017	[22]
		5.9 (n = 118)	Korea	2016–2017	[40]
Phosphonic	Fosfomycin	4.6 (n = 481)	Spain/Lugo	2006–2016	[23]
		2.0 (n = 694)	Austria/Vienna	2016–2018	[49]
		1.9 (n = 161)	Spain/Lugo	2005–2017	[22]
Phenicols	Florfenicol	77.78 (n = 455)	China/Beijing	2014–2016	[42]
		27.9 (n = 129)	China/Tibet	2012	[43]
		40 (n = 55)	United States	2013–2014	[44]
		92.6 (n = 608)	China/Shanghai	2009–2021	[48]
	Chloramphenicol	58.5 (n = 481)	Spain/Lugo	2006–2016	[23]
		18.5 (n = 694)	Austria/Vienna	2016–2018	[49]
		76.61 (n = 455)	China/Beijing	2014–2016	[42]
		57.8 (n = 161)	Spain/Lugo	2005–2017	[22]
		88.1 (n = 118)	Korea	2016–2017	[40]
		16.7 (n = 90)	Denmark	2018–2019	[51]
		1.2 (n = 168)	China/Shenzhen	2009–2014	[46]
Trimethoprim		69.7 (n = 89)	Denmark/Frederiksberg C	2014	[47]
		53.3 (n = 90)	Denmark	2018–2019	[51]
		13.1 (n = 168)	China/Shenzhen	2009–2014	[46]
Folate pathway inhibitors	Trimethoprim-sulfamethoxazole	72.3 (n = 481)	Spain/Lugo	2006–2016	[42]
		49.5 (n = 694)	Austria/Vienna	2016–2018	[49]
		85.55 (n = 455)	China/Beijing	2014–2016	[43]
		75 (n = 135)	Santa Catarina/Brazil	2016–2017	[52]
		19.4 (n = 129)	China/Tibet	2012	[44]
		59.6 (n = 161)	Spain/Lugo	2005–2017	[22]
		56.8 (n = 118)	Korea	2016–2017	[40]
		30.9 (n = 55)	United States	2013–2014	[44]
		13.1 (n = 168)	China/Shenzhen	2009–2014	[45]
Quinolone	Nalidixic acid	60 (n = 481)	Spain/Lugo	2006–2016	[23]
		90.05 (n = 455)	China/Beijing	2014–2016	[42]
		19.4 (n = 129)	China/Tibet	2012	[43]
		87.6 (n = 161)	Spain/Lugo	2005–2017	[22]
		73.91 (n = 23)	Bangladesh/Tangail	2018	[50]
		61.9 (n = 118)	Korea	2016–2017	[40]
		8.9 (n = 90)	Denmark	2018–2019	[51]
		77.4 (n = 168)	China/Shenzhen	2009–2014	[45]
	Levofloxacin	10.8 (n = 481)	Spain/Lugo	2006–2016	[23]
	Norfloxacin	24.6 (n = 118)	Korea	2016–2017	[40]
Monobactam	Aztreonam	2.2 (n = 694)	Austria/Vienna	2016–2018	[49]
		8.1 (n = 161)	Spain/Lugo	2005–2017	[22]
Glycylcyclines	Tigecycline	1.9 (n = 161)	Spain/Lugo	2005–2017	[22]
Quindoxin	Olaquindox	39.77 (n = 455)	China/Beijing	2014–2016	[42]
Polymyxin		20.47 (n = 455)	China/Beijing	2014–2016	[42]
Nitrofurans	Nitrofurantoin	2.34 (n = 455)	China/Beijing	2014–2016	[42]
		9.3 (n = 161)	Spain/Lugo	2005–2017	[22]
	ESBL-producing isolates	10.6 (n = 161)	Spain/Lugo	2005–2017	[22]
	MDR (≥3 categories)	91.3 (n = 161)	Spain/Lugo	2005–2017	[22]
	MDR (≥6 categories)	59 (n = 161)	Spain/Lugo	2005–2017	[22]
Macrolide	Azithromycin	78.26 (n = 23)	Bangladesh/Tangail	2018	[50]
	Erythromycin	47.82 (n = 23)	Bangladesh/Tangail	2018	[50]
	Tilmicosin	100 (n = 55)	United States	2013–2014	[44]
Lincomycin	Clindamycin	100 (n = 55)	United States	2013–2014	[44]
3-MDR	Isolates resistant to penicillin, and cephalosporins, and at least one other class of antibiotics	36.6 (n = 41, farm 1); 32.4 (n = 34, farm 2); 87 (n = 23, farm 3); 95.8 (n = 48, farm 4); 22.4 (n = 58, farm 5); 95.8 (n = 24, farm 6); 90 (n = 10, farm 7)	Germany Mecklenburg–Western Pomerania	2018	[53]
5-MDR	Isolates resistant to penicillin and cephalosporins and at least three other classes of antibiotics	4.9 (n = 41, farm 1); 5.9 (n = 34, farm 2); 17.4 (n = 23, farm 3); 14.6 (n = 48, farm 4); 1.7 (n = 58, farm 5); 8.3 (n = 24, farm 6) 0 (n = 10, farm 7)	Germany Mecklenburg–Western Pomerania	2018	[53]

## 5. Prevalence of AMR-Associated Resistance Genes in ETEC

In swine production, the occurrence of AMR among ETEC has been a longstanding problem [54,55]. It is also noteworthy that the tendency of porcine ETEC to express a multidrug-resistant (MDR) phenotype has increased during the last decade [26,54]. As a result of the continuous use of antimicrobials, it is plausible that at least some MDR ETEC will probably develop pan-resistance, which means a phenotype with resistance to all commonly applied drugs plus resistance to vital antimicrobials such as fluoroquinolones and third- and fourth-generation cephalosporins [56]. It has been demonstrated that the combination of specific virulence-associated plasmids with resistance-associated plasmids is directly related to the fitness of drug-resistant ETEC in the swine production environment [26]. F-type plasmids generally encode for virulence genes, whereas the IncFII-like IncFV, IncA/C, and Incl1 plasmids encode for resistance genes [56,57]. Plasmids encoding for virulent enterotoxins (for example, LT, STb) and the ones associated with antibiotic resistance are commonly transferred together [56,58]. Additionally, antibiotic resistance genes (ARGs) and virulence genes (VGs) were statistically associated with MDR-ETEC isolates from Canada [54]. In addition, a common plasmid harboring both enterotoxin VGs and tetracycline ARG (tetB) was demonstrated in F18-positive O141 [59] and O149:H10 [60] strains [26]. Lastly, for O149:H10 strains, an enhanced virulence was evidenced [60]. Based on these findings, Martínez and Baquero hypothesized that the application of antibiotics may potentially promote the transmission of virulence genes between bacteria [61].

Some studies have established an association between ETEC infection and the presence of associated-antimicrobial resistance genes, as shown in Table 5. According to our analysis, in resistance genes linked to the antimicrobial class of aminoglycosides, there is a heterogeneous distribution at the level of prevalence and type of gene along the countries explored. In Denmark, the most prevalent genes were *aph* and *aadA* with very similar values of 64.4% and 63.3%, respectively [51]. Similarly, Australia showed a similar prevalence for the *aadA* gene with 58.6% [56]. However, in this country, the gene *ant(3)-I* was the most predominant, with 93.3% [26]. For Korea and Switzerland [26], *aac(3)-III* and *strB* were the most common genes. In the study carried out in Korea, Choi et al. [62] evaluated the presence of aminoglycoside-resistant genes in pathogenic and non-pathogenic *E. coli* isolates from pigs during 2004–2007 and concluded that the prevalence of gentamicin/apramycin resistance-associated genes was much higher in diseased pigs than in healthy pigs.

Regarding the class of β-lactams, the gene *bla_TEM_* is very widespread and is associated with a significant prevalence around the world but with different subtypes such as *bla_TEM-1_*, *bla_TEM-1-A_*, *bla_TEM-1-B_*, and *bla_TEM-30_*. This is not surprising given the fact that extended-spectrum β-lactam (ESBL) and AmpC β-lactamase production constitute an important resistance mechanism in members of the *Enterobacteriaceae* family, in which *E. coli* is included [63]. In the literature, IncI1 plasmids are evidenced for the possession of genes encoding for antimicrobial resistance, namely ESBL genes [63,64,65]. Based on this knowledge, Johnson and colleagues [66] evaluated the presence of Incl1-associated genes (*ardA, pill*, and *repl*) and four genes related to β-lactam resistance (*bla_CTX_, bla_CMY-2_, bla_NDM-1_*, and *bla_TEM_*) in ETEC isolates from commercial farms of the United States, corresponding to 88 cases of postweaning diarrhea and 111 cases of neonatal diarrhea. Of the cases examined, 60–66% and 37–40% harbored IncI1 plasmid-associated genes for PWD and ND, respectively. Regarding the β-lactam resistance genes, PWD isolates held *bla_CMY-2_* and *bla_TEM_* at a prevalence of 41% and 50.9%, respectively, whereas the ND isolates held these genes at rates of 25.3% and 22.8%, respectively [66].

Concerning the phenicol class, Switzerland [24] is highlighted with a higher prevalence of *catA1* and *catIII*, 67% and 50% for the respective genes, whereas for Denmark [51] and Australia [26], these genes are expressed at low rates.

Finally, relating to the sulfonamides class, *sul1* and *sul2* genes are the most common for Denmark [51], Switzerland [24], and Australia [26,56], with values above 30% of prevalence.

At a global overview, it is important to note that Australian porcine ETEC was different from isolates of other parts of the world due to the geographical isolation and decades of prohibition of the importation of livestock and fresh meat. For instance, it is very relevant to note that this is a unique country that never permitted the application of fluoroquinolones and gentamicin in food-producing animals [67]. In addition, the usage of third-generation cephalosporins for mass medication is very restricted, and fourth-generation cephalosporins are not registered for application [26,68]. Consequently, these isolates are absent of resistance to the following critically relevant antimicrobials, third-generation cephalosporins, and fluoroquinolones [62].

**Table 5 antibiotics-12-00682-t005:** The prevalence of resistance genes among the ETEC isolates from swine over the world.

Antibiotic Group	Gene	% Prevalence (n = Swine Isolates)	Country/City	Year/Time Range of the Study	Reference
Aminoglycosides	*aph* (phosphotransferases)	64.4% (N = 90)	Denmark	F4-positive isolates: 2018, 2019, and 1989–1992 F18 isolates: 2019 and with a strain recovered in the 1970s	[51]
	*aphA1*	27.1% (N = 70)	Australia	1999–2005	[56]
	*aadA* (nucleotidyltransferases)	63.3% (N = 90)	Denmark	F4-positive isolates: 2018, 2019, and 1989–1992 F18 isolates: 2019 and with a strain recovered in the 1970s	[51]
		58.6% (n = 70)	Australia	1999–2005	[56]
	*aac* (acetyltransferases)	10% (N = 90)	Denmark	F4-positive isolates: 2018, 2019, and 1989–1992 F18 isolates: 2019 and with a strain recovered in the 1970s	[51]
	*aac(3)-II*	18.3% (N = 71) ^a^	Korea	2004–2007	[62]
	*aac(3)-III*	31% (N = 71) ^a^			
	*aac(3)-IV*	47.1% (N = 70)	Australia	1999–2005	[56]
	*ant(2″)-I*	7% (N = 71) ^a^	Korea	2004–2007	[62]
	*armA*	2.8% (N = 71) ^a^			
	*ant(3)-I*	93.3% (N = 104)	Australia	1999–2005	[26]
	*aac(3)-IV*	47.1% (N = 104)			
	*aphA-I*	27.9% (N = 104)			
	*strA*	8% (N = 119) ^b^	Switzerland	2014–2015	[24]
		50% (N = 70)	Australia	1999–2005	[56]
	*strB*	16% (N = 119) ^b^	Switzerland	2014–2015	[24]
		55.7% (N = 70)	Australia	1999–2005	[56]
Beta-lactams	*bla_TEM-1-A_*	4.4% (N = 90)	Denmark	F4-positive isolates: 2018, 2019, and 1989–1992 F18 isolates: 2019 and with a strain recovered in the 1970s	[51]
	*bla_TEM-1-B_*	46.7% (N = 90)			
	*bla_TEM-30_*	1.1% (N = 90)			
	*bla_TEM-1_*	87% (N = 119) ^b^	Switzerland	2014–2015	[24]
	*bla_TEM_*	43.3% (N = 104)	Australia	1999–2005	[26]
		38% (N = 199)	United States	2007–2008	[66]
		40% (N = 70)	Australia	1999–2005	[56]
	*bla_CMY-2_*	34% (N = 199)	United States	2007–2008	[66]
Lincosamides	*Inu*(F)	5.6 % (N = 90)	Denmark	F4-positive isolates: 2018, 2019, and 1989–1992 F18 isolates: 2019 and with a strain recovered in the 1970s	[51]
	*Inu*(G)	5.6 % (N = 90)			
Macrolides	*mdf*(A)	100% (N = 90)	Denmark	F4-positive isolates: 2018, 2019, and 1989–1992 F18 isolates: 2019 and with a strain recovered in the 1970s	[51]
	*mph*(A)	8.9% (N = 90)			
	*mph*(B)	7.8% (N = 90)			
	*erm*(B)	10% (N = 90)			
	*ereA*	7.1% (N = 70)	Australia	1999–2005	[56]
Phenicols	*catA1*	3.3% (N = 90)	Denmark	F4-positive isolates: 2018, 2019, and 1989–1992 F18 isolates: 2019 and with a strain recovered in the 1970s	[51]
		67% (N = 119) ^b^	Switzerland	2014–2015	[24]
	*cmlA1*	8.9% (N = 90)	Denmark	F4-positive isolates: 2018, 2019, and 1989–1992 F18 isolates: 2019 and with a strain recovered in the 1970s	[51]
	*floR*	5.6% (N = 90)			
	*catAIII*	50% (N = 119) ^b^	Switzerland	2014–2015	[24]
	*catI*	9.6% (N = 104)	Australia	1999–2005	[26]
	*catII*	1% (N = 104)			
	*cmlA*	31.7% (N = 104)			
		12.9% (N = 70)	Australia	1999–2005	[56]
Polymyxins	*mcr-1*	26.4% (N = 186)	Spain	2006–2017	[69]
	*mcr-4*	72.8% (N = 186)			
	*mcr-5*	3.6% (N = 186)			
Sulfonamides	*sul1*	33.3 % (N = 90)	Denmark	F4-positive isolates: 2018, 2019, and 1989–1992 F18 isolates: 2019 and with a strain recovered in the 1970s	[51]
		57% (N = 119) ^b^	Switzerland	2014–2015	[24]
		65.4% (N = 104)	Australia	1999–2005	[26]
		57.1% (N = 70)	Australia	1999–2005	[62]
	*sul2*	46.7% (N = 90)	Denmark	F4-positive isolates: 2018, 2019, and 1989–1992 F18 isolates: 2019 and with a strain recovered in the 1970s	[51]
		64% (N = 119) ^b^	Switzerland	2014–2015	[24]
		20.2% (N = 104)	Australia	1999–2005	[26]
		21.4% (N = 70)	Australia	1999–2005	[62]
	*sul3*	10% (N = 90)	Denmark	F4-positive isolates: 2018, 2019, and 1989–1992 F18 isolates: 2019 and with a strain recovered in the 1970s	[51]
		31% (N = 119) ^b^	Switzerland	2014–2015	[24]
Tetracycline	*tet*(A)	44.4% (N = 90)	Denmark	F4-positive isolates: 2018, 2019, and 1989–1992 F18 isolates: 2019 and with a strain recovered in the 1970s	[51]
		65% (N = 119) ^b^	Switzerland	2014–2015	[24]
		44.2% (N = 104)	Australia	1999–2005	[26]
		35.7% (N = 70)	Australia	1999–2005	[56]
	*tet*(B)	14.4% (N = 90)	Denmark	F4-positive isolates: 2018, 2019, and 1989–1992 F18 isolates: 2019 and with a strain recovered in the 1970s	[51]
		23% (N = 119) ^b^	Switzerland	2014–2015	[24]
		28.8% (N = 104)	Australia	1999–2005	[26]
		7.1% (N = 70)	Australia	1999–2005	[56]
	*tet(C)*	35% (N = 119) ^b^	Switzerland	2014–2015	[24]
		16.3% (N = 104)	Australia	1999–2005	[26]
		5.7% (N = 70)	Australia	1999–2005	[56]
	*tet(D)*	3% (N = 119) ^b^	Switzerland	2014–2015	[24]
	*tet(E)*	2% (N = 119) ^b^	Switzerland	2014–2015	[24]
	*tet*(X)	1.1% (N = 90)	Denmark	F4-positive isolates: 2018, 2019, and 1989–1992 F18 isolates: 2019 and with a strain recovered in the 1970s	[51]
Trimethoprim	*dfr*A1	37.8% (N = 90)	Denmark	F4-positive isolates: 2018, 2019, and 1989–1992 F18 isolates: 2019 and with a strain recovered in the 1970s	[51]
		59% (N = 119) ^b^	Switzerland	2014–2015	[24]
	*dfr*A5	2.2% (N = 90)	Denmark	F4-positive isolates: 2018, 2019, and 1989–1992 F18 isolates: 2019 and with a strain recovered in the 1970s	[51]
		7% (N = 119) ^b^	Switzerland	2014–2015	[24]
	*dfrA7*	7% (N = 119) ^b^	Switzerland	2014–2015	[24]
	*dfr*A12	8.9% (N = 90)	Denmark	F4-positive isolates: 2018, 2019, and 1989–1992 F18 isolates: 2019 and with a strain recovered in the 1970s	[51]
		10% (N = 119) ^b^	Switzerland	2014–2015	[24]
	*dfrA13*	7% (N = 119) ^b^	Switzerland	2014–2015	[24]
	*dfr*A14	5.6% (N = 90)	Denmark	F4-positive isolates: 2018, 2019, and 1989–1992 F18 isolates: 2019 and with a strain recovered in the 1970s	[51]
		5% (N = 119) ^b^	Switzerland	2014–2015	[24]
	*df*rA17	2.2% (N = 90)	Denmark	F4-positive isolates: 2018, 2019, and 1989–1992 F18 isolates: 2019 and with a strain recovered in the 1970s	[51]
		5% (N = 119) ^b^	Switzerland	2014–2015	[24]
	*dfrA19*	7% (N = 119) ^b^	Switzerland	2014–2015	[24]
	*dhfrI*	1.9% (N = 104)	Australia	1999–2005	[26]
	*dhfrV*	31.7% (N = 104)	Australia	1999–2005	[26]
		25.7% (N = 70)	Australia	1999–2005	[56]
	*dhfrXIII*	30.8% (N = 104)	Australia	1999–2005	[26]

^a^ Pathogenic *E.coli* isolates; ^b^ enterovirulent *E. coli* from pigs (where N_ETEC_ = 66).

### Horizontal Gene Transfer

In addition to the identification of ETEC-associated AMR genes and their prevalence, it is demanding to understand how the ETEC-associated VGs emerge and disseminate across species since both classes of genes (VGs, AMR) are intrinsically related, as was explained above. It has been described that variability in VG and colonization factor combinations highlight the genomic diversity within the ETEC pathogroup [70]. These data suggest that ETEC consists of a genetically heterogeneous group of strains that gained the ETEC-associated virulence genes by horizontal gene transfer. In fact, it has been shown that strains within a single pathogroup can originate from distinct genetical backgrounds [70,71,72,73]. Multi-locus sequence typing (MLST) has shown that ETEC strains originate from different evolutionary lineages, proposing that the acquisition of the *elt* or *est* genes may be enough to make an ETEC strain [74]. This hypothesis is also supported by the study carried out by Chen and colleagues, which shows that the prototypical ETEC strain H10407 chromosome is almost identical to the chromosome of *E*. *coli* K-12 strain MG1655, suggesting that the major event in the emergence of ETEC from *E*. *coli* is, thus, the acquisition of virulence plasmids carrying *elt* or *est* [75]. Contrary to these current notions, recent evidence, based on the sequence analysis of a representative collection of isolates of ETEC isolated between 1980 and 2011, showed that persistent plasmid-chromosomal background combinations exist in certain phylogenetic lineages [76]. Due to these divergences, additional research is needed to understand the gene transfer between strains to improve the prevention as well as treatment. As such, a more detailed understanding of what actually constitutes a naturally occurring ETEC strain is vital.

## 6. Conclusions

It is indispensable to control the application of antimicrobials to treat ETEC-associated infections in swine farms in order to reduce the incidence of AMR and, consequently, the probability of zoonoses occurrence, which implies a public health concern derivate from the potential transfer of AMR genetic determinants directly by contact and indirectly into the food chain, water, manure, and others [5]. It is crucial to note that high levels of AMR in ETEC strains have been arising, namely in apramycin, neomycin, sulfonamide-trimethoprim, and colistin.

In Europe, the EFSA AHAW Panel (2021c) revealed clinical swine *E. coli* isolates with a high proportion of resistance to numerous antibiotics with a prevalence from 63% to 70% (namely to aminopenicillins, sulfonamides, and tetracycline) [77]. However, lower rates of resistance to clinically critical antibiotics (fluoroquinolones and third-generation cephalosporins) were detected. Although, the risk of development of a pan-resistant MDR ETEC remains and may constitute a huge problem in public health. Therefore, alternative therapeutics (prebiotics, probiotics, synbiotics, organic acids, phytogenic substances, bacteriophages, specific egg yolk antibodies, lactoferrin, antisense oligonucleotides, spray-dried animal plasma, and aptamers), as well as hygienic and sanitary measurements, should be applied in pig farming [2,4].

In sum, this knowledge about the distribution of pathogenic ETEC in swine farms and their diversity, resistance, and virulence profiles constitute a preliminary measure to adopt the best treatments. However, this review has some limitations, the assembled data can be outdated since the mutagenic capacity of *E. coli* constitutes an important aspect to take into account. In addition, some countries do not possess studies in the scope of this review. Therefore, from a future research perspective, it is crucial to evaluate alternative therapies in vitro and in vivo capable of decreasing the usage of antimicrobials, characterize serotypes of ETEC in Portugal since this information is scarce, and combine bioinformatic tools to complement the genetic composition characterization.

## Figures and Tables

**Figure 1 antibiotics-12-00682-f001:**
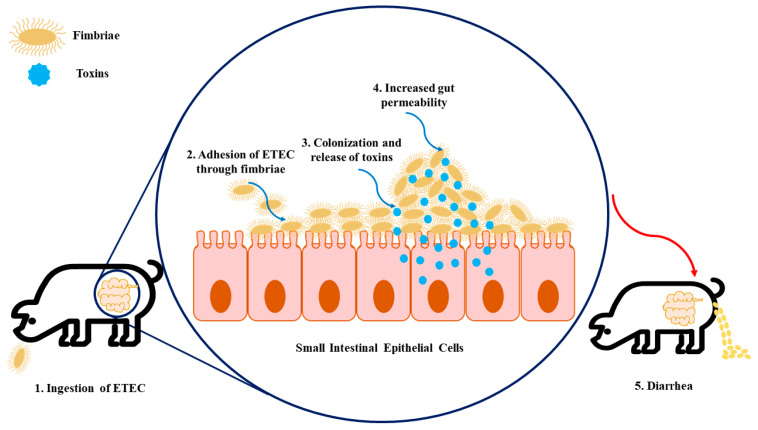
The infection model of enterotoxigenic *Escherichia coli* (ETEC) on intestinal epithelial cells. (**1**) Firstly, the swine ingest ETEC, enabling its transition to the gastrointestinal tract. (**2**) The fimbriae expressed by ETEC allow the adhesion of bacteria to specific receptors present in the intestinal epithelial cells. (**3**) Colonization arises in the small intestinal mucosa, which leads to the production of toxins. (**4**) These enterotoxins promote water and electrolyte loss into the intestinal lumen, resulting in increased gut permeability. (**5**) As a consequence of increased gut permeability and massive water loss, diarrhea, weight loss, and mortality can happen.

**Table 2 antibiotics-12-00682-t002:** Prevalence of ETEC in several countries from studies published since 2010.

Country	Prevalence of ETEC (%) (n = Number of Isolates)	Sampling Information/Origin	Period	Reference
Argentina	15.2 (n = 990)	11 farms with no history or clinical signs of colibacillosis	2015	[16]
Australia	58.8 (n = 325)	22 pig herds	2013–2014	[17]
Belgium and the Netherlands	36.4 (n = 160)	88 farms	2012–2014	[18]
France	64.8 (n = 455)	91 farms	2012–2014	[18]
Germany	47.1 (n = 99)	17 farms	2012–2014	[18]
Italy	81.0 (n = 159)	84 farms	2012–2014	[18]
Poland	30 (n = 386) ^a^	70 pig herds	2011–2013	[19]
South Africa	72.0 (n = 228)	8 piggeries of different sizes (16–650 sow units) and production systems: large-scale commercial (>250 sow units), medium-scale commercial (51–250 sow units), and emerging small-scale pig farms (<50 sow units)	2015–2016	[20]
South Africa	18.6 (n = 263)	263 neonatal and post-weaned pigs	2013	[21]
Spain	86.5 (n = 186)	50 different Spanish farms	2005–2017	[22]
Spain	67.0 (n = 499)	179 outbreaks	2008–2018	[23]
Switzerland	50.4 (n = 131)	115 pigs suffering from diarrhea	2014–2015	[24]

^a^ Prevalence of enterotoxigenic *E coli* with fimbriae F4 (ETEC-F4).

## Data Availability

Not applicable.

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
