# Peer review of "Swine Colibacillosis: Global Epidemiologic and Antimicrobial Scenario"

_antibiotics, 2023, doi:10.3390/antibiotics12040682_

Round 1

Reviewer 1 Report

Comments to the Authors

At the outset, I would like to thank you for the opportunity to participate in the review of the manuscript entitled (Swine colibacillosis: global epidemiologic and antimicrobial scenario) The current review article studied the Swine colibacillosis caused by pathogenic Escherichia coli that represents an epidemiological challenge not only for animal husbandry but also for health authorities. This review article explores the distribution of pathogenic ETEC in swine farms, their diversity, resistance, and virulence profiles, and finally, summarizes the most relevant works on these subjects over the past 10 years and discusses the importance of these bacteria as zoonotic agents.

The abstract as well as the introduction to the article is clearly written, introduces the topic well, and the selection of literature is appropriate.

The purpose and scope of the article have been correctly defined. The figures and tables are clearly described, and the discussions are well presented.

I consider the entire manuscript interesting and worthy of attention. The manuscript was pleasant to read.

Overall, nice work. Congratulations.

Author Response

First of all, we would like to express our appreciation to the reviewer for their careful reading of the text.

The revised manuscript has all changes highlighted in green to allow better follow-up by the reviewer.

REVIEWER 1

Comments to the Authors

“At the outset, I would like to thank you for the opportunity to participate in the review of the manuscript entitled (Swine colibacillosis: global epidemiologic and antimicrobial scenario) The current review article studied the Swine colibacillosis caused by pathogenic Escherichia coli that represents an epidemiological challenge not only for animal husbandry but also for health authorities. This review article explores the distribution of pathogenic ETEC in swine farms, their diversity, resistance, and virulence profiles, and finally, summarizes the most relevant works on these subjects over the past 10 years and discusses the importance of these bacteria as zoonotic agents.

The abstract as well as the introduction to the article is clearly written, introduces the topic well, and the selection of literature is appropriate.

The purpose and scope of the article have been correctly defined. The figures and tables are clearly described, and the discussions are well presented.

I consider the entire manuscript interesting and worthy of attention. The manuscript was pleasant to read.

Overall, nice work. Congratulations.”

Authors answer: We are pleased with the congratulation and for the perception of the relevance of the topic of this review.

Reviewer 2 Report

Manuscript ID: antibiotics-2306014-peer-review-v1

The topic “Swine colibacillosis: global epidemiologic and antimicrobial scenario” This manuscript is interesting in the field. However, before accepting for publication, some point of the manuscript needs to revise and make clear for correct understanding.

Comments are below…

1. Typographical errors exist throughout the manuscript. Rectify carefully.

2. The objective of this review is still not clear, try to revise it for relating to the detail. 

3. The plagiarism was detected at a high level, please modify it based on the evident file herewith.

4. The research article addresses on the antibiotic used in the swine, if they had to enter the biological systems what will be the ways of elimination or the fate of disposal from the biological food chain? Please Justify.

5. In general, the EHEC may promote a negative effect on the swine or mammalian, so, what is the reason behind that author did not mention this pathogen.?

6. Check Table 1. The number and letter were collapsed.

7. In the conclusion seems to be in general and is not given separately, it is highly recommended to include limitations of the study and potential future research goals.

Author Response

First of all, we would like to express our appreciation to the reviewer for their careful reading of the text and for all the in-depth constructive comments and suggestions about the first version of our manuscript. We have put an earnest effort to respond to the concerns of each reviewer in detail and we are now submitting a revised manuscript.

We are confident that the changes performed are in accordance with the reviewer's requests and hope the reviewer feels that our manuscript is now acceptable for publication.

A point-by-point description of our answers to the reviewer’s comments and suggestions follows. The revised manuscript has all changes highlighted in green to allow better follow-up by the reviewer.

REVIEWER 2

“The topic “Swine colibacillosis: global epidemiologic and antimicrobial scenario” This manuscript is interesting in the field. However, before accepting for publication, some point of the manuscript needs to revise and make clear for correct understanding.

Comments are below…”

  1. “Typographical errors exist throughout the manuscript. Rectify carefully.”

Authors answer: We have corrected the typographical errors in the revised version of the manuscript.

“2. The objective of this review is still not clear, try to revise it for relating to the detail.”

Authors answer: We are in concordance with this statement, and we include a paragraph at the end of the introduction with the main goal of this review in detail.

“3. The plagiarism was detected at a high level, please modify it based on the evident file herewith.”

Authors answer: We carefully rectified the feasible similar terms in this review, when compared with published literature.

“4. The research article addresses on the antibiotic used in the swine, if they had to enter the biological systems what will be the ways of elimination or the fate of disposal from the biological food chain? Please Justify.”

Author answers: We appreciate the reviewer's question and we pretend to explain this topic minutely.

As well pointed out by the reviewer, although all the efforts that have been conducted to limit the use of antibiotics, can enter the environment/biological system through different routes.

In fact, antibiotics used in the farm industry as well as their metabolites can be released from pigs through feaces. Variable waste streams typical of industrial production will likely require a range of treatment technologies. A major challenge is that the high antibiotic concentrations in industrial wastewater treatment plants (WWTPs) inevitably will exert a strong selection for antibiotic resistance bacteria (ARBs). For this reason, according to Pruden and colleagues [1], activated sludge is not recommended for highly antibiotic-contaminated waste streams because of the high density of microbial populations. If biological treatment is unavoidable, bacteria from the treatment process must be eliminated before discharge. Of note, if swine farm’ wastewater treatment processes are not effectively treated, they can threaten animal production, public health and the ecological safety of the surrounding environment.

On the other hand, the farm-to-fork continuum was highlighted as a possible reservoir of AMR, and a hotspot for the emergence and spread [2-3] of AMR.  Trying to overcome somewhat of the issues associated with the use of antibiotics and meat safety, the use of antimicrobials for the growth promotion of food animals has been banned in several countries. As we also reported in the manuscript, it is noteworthy, that in Europe, according to Regulation (EU) 2019/61 on Veterinary Medicines and Regulation (EU) 2019/4 on Medicated Feed, antibiotics shall not be applied routinely, neither used for prophylaxis, unless in exceptional cases. Similarly, in the USA, since 2017, only therapeutic use (treatment, control, prevention) for a specific animal health condition is allowed under the direction of a veterinarian. It is important to note, that when an animal is treated with antibiotics, farmers must follow the guidelines of the country defining how long they must withhold that animal’s meat consumption.

In addition, antimicrobial resistance in the food chain can be also influenced by several factors, such as the horizontal ARGs transfer or the induction of AMR related to responses directed to diverse food chain stresses, namely by sublethal exposure or recurring exposure to: (i) an extensive diversity of antimicrobial agents, for example, the disinfectants used in several steps of the food chain, but also agrochemicals or even food preservatives; (ii) different physical treatments used in food processing, such as thermal and non-thermal treatments; (iii) new food safety approaches, such as the use of bacteriophage as an alternative to antibiotics in animal health or as biopreservatives.

In sum, an effective monitorization of antibiotic therapy in farm animals is essential for reducing antimicrobial resistance and, in its turn, enhancing the safety of food. Furthermore, the co-selection or cross-adaptation phenomena of bacteria need further study, due to their complexity and relevance as a public health concern [2].

References used to support this comment:

[1] Pruden A.; Larsson D. G.; Amézquita A.; Collignon P.; Brandt K. K.; Graham D. W.; Lazorchak J. M.; Suzuki S.; Silley P.; Snape J. R.; Topp E.; Zhang T.; Zhu Y. G. Management options for reducing the release of antibiotics and antibiotic resistance genes to the environment. Environ Health Perspect 2013, 121, 878-885.

[2] Giacometti F.; Shirzad-Aski H.; Ferreira S. Antimicrobials and Food-Related Stresses as Selective Factors for Antibiotic Resistance along the Farm to Fork Continuum. Antibiotics (Basel) 2021, 10, 671, 1-29.

[3] Kumar S. B.; Arnipalli S. R.; Ziouzenkova O. Antibiotics in Food Chain: The Consequences for Antibiotic Resistance. Antibiotics (Basel) 2020, 9, 688, 1-26.

“5. In general, the EHEC may promote a negative effect on the swine or mammalian, so, what is the reason behind that author did not mention this pathogen?”

Author answers: We understand the doubt regarding the negative effect of EHEC on swine or mammalian and we appreciate the recommendation. As such, we have changed the manuscript,  since, in the previous version, we had this pathotype included in their main group, the STEC. But now we include the subtypes of this group, namely EDEC and EHEC, and we have added yet their relation with the attaching and effacing (A/E) lesion development. Behind this, since the EPEC group also allows this A/E lesion, we have complemented this information too. However, both these pathotypes (EPEC, EHEC) are less pathogenic to swine, especially the EHEC one. Conversely, this EHEC pathotype is highly pathogenic to humans and has relevance as a zoonotic agent. Concerning EPEC, some bacteria are recovered in cases of PWD.  

“6. Check Table 1. The number and letter were collapsed.”

Author answers: We attend to the reconfiguration of table 1.

“7. In the conclusion seems to be in general and is not given separately, it is highly recommended to include limitations of the study and potential future research goals.”

Author answers: We agree with these suggestions. Thus, we have included some limitations aspects as well as the potential future research goals intended.

Reviewer 3 Report

The topic of this paper was relevant, timely, and of interest to the audience of this journal.

The content of this paper was technically accurate and sound.

Author Response

First of all, we would like to express our appreciation to the reviewer for their careful reading of the text.

The revised manuscript has all changes highlighted in green to allow better follow-up by the reviewer.

REVIEWER 3

“The topic of this paper was relevant, timely, and of interest to the audience of this journal. The content of this paper was technically accurate and sound.”

 Author answers: We are flattered by all these remarks.

Reviewer 4 Report

 The article entitled “Swine colibacillosis: global epidemiologic and antimicrobial scenario” by Maria Margarida Barros et al.

It’s a very detailed review article.

The idea is very good and innovative.

Introduction is very well written and well explained.

On each aspect the article is very well elaborated with perfect information.

I would recommend acceptance of the manuscript but there are some grammatical mistakes which the authors should correct them

Author Response

First of all, we would like to express our appreciation to the reviewer for their careful reading of the text.

We are confident that the changes performed are in accordance with the reviewer's requests and hope the reviewer feels that our manuscript is now acceptable for publication.

A point-by-point description of our answers to the reviewer’s comments and suggestions follows. The revised manuscript has all changes highlighted in green to allow better follow-up by the reviewer.

REVIEWER 4

“The article entitled “Swine colibacillosis: global epidemiologic and antimicrobial scenario” by Maria Margarida Barros et al.

It’s a very detailed review article.

The idea is very good and innovative.

Introduction is very well written and well explained.

On each aspect the article is very well elaborated with perfect information.

I would recommend acceptance of the manuscript but there are some grammatical mistakes which the authors should correct them.”

Author answers: We are grateful for all these comments and we have attended to all the grammatical mistakes.

Round 2

Reviewer 2 Report

The revised version was clear point by point. So, this manuscript is OK for acceptance and publication.